# Hydrogen Sulfide Treatment Improves Post-Infarct Remodeling and Long-Term Cardiac Function in CSE Knockout and Wild-Type Mice

**DOI:** 10.3390/ijms21124284

**Published:** 2020-06-16

**Authors:** Leigh J. Ellmers, Evelyn M. Templeton, Anna P. Pilbrow, Chris Frampton, Isao Ishii, Philip K. Moore, Madhav Bhatia, A. Mark Richards, Vicky A. Cameron

**Affiliations:** 1Christchurch Heart Institute, Department of Medicine, University of Otago, Christchurch 8140, New Zealand; ellmersleigh@gmail.com (L.J.E.); evie.templeton@postgrad.otago.ac.nz (E.M.T.); anna.pilbrow@otago.ac.nz (A.P.P.); Statistecol@xtra.co.nz (C.F.); mark.richards@nus.edu.sg (A.M.R.); 2Laboratory of Health Chemistry, Showa Pharmaceutical University, Tokyo 194-8543, Japan; isao-ishii@umin.ac.jp; 3Department of Pharmacology, National University of Singapore, Singapore 119228, Singapore; dprmpk@nus.edu.sg; 4Inflammation Research Group, Department of Pathology and Biomedical Science, University of Otago, Christchurch 8140, New Zealand; madhav.bhatia@otago.ac.nz; 5Cardiovascular Research Institute, National University of Singapore, Singapore 119228, Singapore

**Keywords:** hydrogen sulfide, myocardial infarction, knockout mice, gene expression

## Abstract

Hydrogen sulfide (H_2_S) is recognized as an endogenous gaseous signaling molecule generated by cystathionine γ-lyase (CSE) in cardiovascular tissues. H_2_S up-regulation has been shown to reduce ischemic injury, and H_2_S donors are cardioprotective in rodent models when administered concurrent with myocardial ischemia. We evaluated the potential utility of H_2_S therapy in ameliorating cardiac remodeling with administration delayed until 2 h post-infarction in mice with or without cystathionine γ-lyase gene deletion (CSE^−/−^). The slow-release H_2_S donor, GYY4137, was administered from 2 h after surgery and daily for 28 days following myocardial infarction (MI) induced by coronary artery ligation, comparing responses in CSE^−/−^ with wild-type (WT) mice (*n* = 5–10/group/genotype). Measures of cardiac function and expression of key genes associated with cardiac hypertrophy, fibrosis, and apoptosis were documented in atria, ventricle, and kidney tissues. Post-MI GYY4137 administration reduced infarct area and restored cardiac function, accompanied by reduction of the elevated ventricular expression of genes mediating cardiac remodeling to near-normal levels. Few differences between WT and CSE^−/−^ mice were observed, except CSE^−/−^ mice had higher blood pressures, and higher atrial *Mir21a* expression across all treatment groups. These findings suggest endogenous CSE gene deletion does not substantially exacerbate the long-term response to MI. Moreover, the H_2_S donor GYY4137 administered after onset of MI preserves cardiac function and protects against adverse cardiac remodeling in both WT and CSE-deficient mice.

## 1. Introduction

Hydrogen sulfide (H_2_S) is a colorless, flammable, toxic gas, which is now recognized to act as an endogenous gaseous signaling molecule across a range of tissue systems [1,2]. Numerous studies have demonstrated that H_2_S provides protection against cellular injury after ischemia-reperfusion in brain, liver, lung, kidney, and heart [1]. Accumulating evidence suggests that in myocardial ischemia, the up-regulation of endogenous H_2_S or treatment with H_2_S donors reduces the extent of cardiac ischemic injury [3,4,5,6], decreasing mortality, improving left ventricular pressures, suppressing leukocyte infiltration, either increasing or reducing pro-inflammatory cytokines (depending on the H_2_S donor used), and attenuating fibroblast hyperplasia [7]. Similarly, H_2_S treatment has been demonstrated to diminish the transition from pressure overload hypertrophy to heart failure in mice [8] and to slow the progression of cardiac remodeling and promote angiogenesis in a mouse model of congestive heart failure [9]. Hence, new therapeutic compounds with the ability to deliver H_2_S may provide an attractive approach in treating cardiovascular disease [2].

A broad range of cellular and molecular signaling pathways mediate the cardioprotective effects of H_2_S within the cardiovascular system. These include covalent modification of target protein cysteine residues, activation of sarcolemma ATP-sensitive K channels (K_ATP_), and activation of signaling pathways mediated by PI3K/Akt, PKC, extracellular regulated kinase 1/2 (ERK1/2) [10], signal transducers, and activators of transcription-3 (STAT-3), p90RSK, Bax/Bcl-2 signaling, and heat shock proteins (HSPs). In addition, H_2_S is reported to inhibit the renin-angiotensin system, to have antioxidant effects, and to activate anti-apoptotic signaling [11,12]. Cross talk with nitric oxide (NO) may also contribute to the actions of H_2_S [13,14], inhibiting apoptosis either directly or through inhibiting activation of caspase-3. Finally, the actions of H_2_S to attenuate cardiac injury in mice models may be moderated in part via the induction of *Mir21a* [15], despite previous reports that *Mir21a* may contribute to the atrial fibrotic remodeling following myocardial infarction (MI) [16]. 

Generation of endogenous H_2_S is attributed to three key enzymes; cystathionine γ-lyase (CSE) [17], cystathionine γ-synthase (CBS), and 3-mercaptopyruvate sulfurtransferase (3-MST) [18]. The enzyme CSE is abundant in the heart, liver, kidney, and vascular smooth muscle, and is the most relevant H_2_S-producing enzyme in the cardiovascular system [17,19]. Suppression of endogenous H_2_S levels by pharmacological CSE inhibition following cardiac ischemia/reperfusion increases the extent of myocardial damage in rats [20]. Mice with global CSE gene deletion (CSE^−/−^) show a significant reduction in H_2_S bioavailability [21,22], and greater cardiac dilatation and exacerbated dysfunction in heart failure compared to wild-type mice [5]. Conversely, cardiac-specific CSE-overexpressing transgenic mice subjected to ischemia/reperfusion exhibit decreased infarct size and improved cardiac contractility [23]. Together, these findings point to a role for endogenous H_2_S production via CSE in cardio-protection after ischemia. However, how the deletion of endogenous CSE generation affects long-term cardiac remodeling in response to MI has not previously been documented.

Intriguingly, H_2_S has also been proposed to play a dual protective effect in the cardiorenal syndrome [2], a term for the cycle of acute kidney injury associated with MI or acute decompensated heart failure, with worsening renal function precipitating further cardiac dysfunction [24]. Renal or cardiac fibrosis developing as a result of either kidney damage or cardiac injury respectively, is mitigated by H_2_S, likely acting through TGFβ signaling, with anti-inflammatory and antioxidant effects thought to be via p38 MAPK, ERK1/2, Akt, and Nrf2 pathways, as described above [25]. These findings indicate that H_2_S could have multi-organ protective actions in the cardiorenal syndrome, but the kidney response to myocardial injury has not been previously investigated in the presence of either H_2_S donor drugs or CSE gene deletion, to our knowledge. 

We hypothesized firstly, that H_2_S therapy would have potential utility in ameliorating cardiac remodeling post-infarction in a mouse model, even when administered in a feasible therapeutic time window after the onset of MI. We investigated the effects of treatment with the synthetic, slow-release, H_2_S donor compound, GYY4137 [26], administered from 2 h after MI induced via permanent coronary artery ligation and daily during 28-days recovery following surgery, documenting indices of cardiac hypertrophy and cardiac function. In addition, we hypothesized that mice with global CSE gene deletion would have exacerbated cardiac injury and impaired cardiac function post-MI compared with the corresponding wild-type mice (WT). Therefore, to elucidate long-term physiological and molecular effects of endogenous (CSE-generated) H_2_S, we examined the response to MI in CSE^−/−^ mice [27] compared to WT mice. To elucidate signaling pathways involved in the responses to MI and CSE gene deletion, separately and in combination, we documented expression of selected genes associated with cardiac hypertrophy (ventricular *Nppa*, *Nppa*, *Myh7*, and *Myh6)*, cardiac remodeling and fibrosis (atrial *MiR21a*, ventricular *Tgfb1*, *Gata4*, *Col1a1*, *Nrf2*, *Agtr1*, and *Ace*), and antioxidant and apoptosis pathways genes (*Casp3* and* Akt1*). Finally, because of the dual protective effects of H_2_S on both heart and kidney [2], we hypothesized that H_2_S would extend renal protection in this model, and investigated the kidney expression of genes involved in hypertrophic, fibrotic, and apoptotic pathways. Our findings demonstrated that, contrary to our second hypothesis, endogenous CSE gene deletion did not substantially exacerbate the long-term response to MI. Moreover, in both WT and CSE-deficient mice, the H_2_S donor GYY4137 administered after onset of MI preserved cardiac function and protected against adverse cardiac remodeling.

## 2. Results

### 2.1. Infarct Size, Blood Pressure, Heart Weight/Body Weight, Cardiac Function Post-MI

All mice in both Sham and MI groups survived the full study period of 28 days. Measures of blood pressure, infarct area at end of study, cardiac function, and echocardiography indices of mass and volume are presented in Table 1. Consistent with prior reports, CSE^−/−^ mice had higher levels of mean arterial pressure (MAP, recorded under light anesthesia) at end of study (*p =* 0.001). Infarction had profound effects on most measures of cardiac function and of adverse ventricular remodeling, while GYY4137 drug treatment administered over the 28 days post-MI period was associated with a significant amelioration of these effects. The area of infarct expansion after 28 days was significantly smaller with GYY4137 treatment compared to the vehicle infarct groups in both WT mice (*p* = 0.005) and CSE^−/−^ mice (*p* = 0.009) (Table 1 and Figure 1). In parallel, while significantly higher heart weight/body weight (HW/BW) was observed at 28 days post-MI compared to non-infarcted WT and CSE^−/−^ mice (*p* < 0.001 for both), this was significantly ameliorated with GYY4137 treatment (*p* < 0.001) in both genotypes. MI markedly reduced left ventricular ejection fraction (LVEF) and fractional shortening (FS) in both WT and CSE^−/−^ mice (all *p* < 0.001), and administration of GYY4137 with MI restored both LVEF and FS by the end of the study to equivalent levels of the respective non-infarcted groups (*p* < 0.001 for infarction with and without GYY4137). Only slight differences in cardiac function measures between WT and CSE^−/−^ mice were observed, with lower LVEF post-infarction in CSE^−/−^ (*p* = 0.002). 

Echocardiographic measures (Table 1) of left ventricular (LV) mass varied between individuals and no overall significant differences between groups was observed, notwithstanding the difference in HW/BW. However, systolic LV posterior wall thickness at (LVPWS) was different between groups (*p* < 0.001 overall), showing a decrease with MI by end of study through infarct expansion and wall thinning in both WT (*p* < 0.025) and CSE^−/−^ mice (*p* < 0.001), and a return to control dimensions when MI was followed by GYY4137 administration. In contrast, the interventricular septum dimension at systole (IVSs) was increased with infarct in WT (*p* < 0.001) and decreased to non-infarcted levels with GYY4137 administration. For cardiac volumes, neither end systolic volume (ESV) nor end diastolic volume (EDV) differed across groups, but these are shown in Table 1 for reference. Finally, GYY4137 administration alone in non-infarcted mice had minimal effects on most echocardiographic variables in both genotypes. However, we did observe differential effects of the drug on FS between the two genotypes (*p* = 0.001 for interaction), GYY4137 associated with reduced FS in WT, and increased FS in CSE^−/−^ mice. 

### 2.2. Expression of Atrial MiR21a and Ventricular Hypertrophy, Fibrosis, and Apoptosis Genes

Atrial *MiR21a* expression (Figure 1) was significantly increased at 28 days in response to infarction compared with the respective sham group in both WT (*p* = 0.034) and CSE^−/−^ mice (*p* < 0.05). However, there was no significant effect of GYY4137 administration on atrial *MiR21a* expression, either in the sham or MI groups of either genotype. Paradoxically, CSE^−/−^ mice overall had markedly higher *MiR21a* expression levels than their WT comparator groups (*p* < 0.001 between genotypes overall), with a pairwise difference between genotypes noted even in the baseline condition of non-infarcted mice without drug administration (*p* = 0.002). 

In left ventricular tissue, increased expression of genes associated with cardiac hypertrophy was observed at 28 days after MI (Figure 1, for complete gene expression data see Appendix A), notably for *Nppa* (significant in WT only, *p* = 0.014), *Myh7* (WT *p* = 0.002; CSE^−/−^
*p* = 0.049), and *Myh6* (significant in WT only, *p* = 0.041). However, when MI was combined with GYY4137 administration, the elevated expression levels of these genes post-MI was reduced, often to levels close to those observed in control, non-infarcted mice. Administration of GYY4137 post-MI decreased the elevated expression from MI plus vehicle of *Nppa* (WT *p* = 0.017; CSE^−/−^
*p* = 0.002), *Nppb* (WT *p* = 0.003; ns in CSE^−/−^ mice), *Myh7* (WT *p* = 0.0001; CSE^−/−^
*p* = 0.035), and *Myh6* (CSE^−/−^ mice only *p* = 0.034). In sham, non-infarcted mice, there were no significant effects of GYY4137 drug alone on expression of any of these genes, nor any overall differences between the genotypes.

Among genes associated with cardiac remodeling (Figure 2), elevated ventricular expression was observed after infarction compared to sham for *Tgfb1* (WT *p <* 0.001; CSE^−/−^
*p* < 0.001), but the rise was not significant for *Gata4* expression. However, elevated expression after MI was restored to background levels by GYY4137 drug, being significantly lower with MI plus drug compared to their respective MI plus vehicle group in both *Tgfb1* (WT *p* < 0.001; CSE^−/−^
*p* < 0.001) and *Gata4* (WT *p* = 0.007; CSE^−/−^
*p* = 0.004). Administration of GYY4137 alone in non-infarcted mice had no significant effect on *Tgfb1* or *Gata4* expression, nor were there any differences between the genotypes. 

Of fibrosis-associated genes (Figure 2), elevated levels of expression were observed after infarct compared to sham for *Col1a1* (WT *p* = 0.001; CSE^−/−^
*p* = 0.012) and *Nrf2* (CSE^−/−^
*p* = 0.041, noting that there was no WT infarct group in the *Nrf2* analysis). When GYY4137 was administered post-MI, the elevated expression returned to control levels for *Col1a1* (WT *p* = 0.008; CSE^−/−^
*p* = 0.015). In non-infarcted sham animals, there was no significant effect of GYY4137 treatment alone on *Col1a1* expression, while GYY4137 increased *Nrf2* expression compared to vehicle in both WT (*p* = 0.027) and CSE^−/−^ mice (*p* = 0.019). There were also no differences between the genotypes on *Col1a1* expression, while for *Nrf2*, CSE^−/−^ had lower levels of expression across all groups (*p* = 0.013).

Genes of the renin-angiotensin system (Figure 2) have also been demonstrated to be associated with cardiac remodeling and fibrosis. Elevated expression was observed in response to MI for *Agtr1* (WT *p* = 0.019; CSE^−/−^
*p* = 0.006) and *Ace* (WT *p* < 0.001; CSE^−/−^
*p* = 0.04). Again, GYY4137 administration restored the elevated expression post-MI to control levels for both *Agtr1* (WT *p* = 0.011; CSE^−/−^*p* = 0.003) and *Ace* (WT *p* < 0.001; CSE^−/−^
*p* = 0.002), comparing MI with and without GYY4137. In sham, non-infarcted mice, GYY4137 administration did not alter expression of either *Agtr1* or *Ace*. However, while *Agtr1* did not differ between genotypes, CSE^−/−^ mice had lower *Ace* expression levels relative to their comparator WT mice (overall *p* = 0.013), especially comparing the two genotypes after infarct (*p* = 0.005). For *Agt* expression (Appendix A), there were no significant differences between any of the groups, comparing sham and infarct, vehicle and drug, and no difference between genotypes.

Expression of the antioxidant and apoptosis-associated genes (Figure 3) was increased with infarction compared with sham in WT mice for both *Casp3* (*p* = 0.002) and *Akt1* (*p* = 0.019), but not in CSE^−/−^ mice. Treatment with GYY4137 post-MI restored *Casp3* expression to control levels in both WT (*p* = 0.012) and in CSE^−/−^ (*p* = 0.004), but post-MI *Akt1* expression levels tended to be reduced by GYY4137 significantly in CSE^−/−^ mice only (*p* = 0.009). In non-infarcted sham mice, there were no significant effects of GYY4137 drug alone on *Casp3* expression, but there was a small increase in *Akt1* in CSE^−/−^ mice only (*p* = 0.039). Comparing the genotypes, the *Casp3* response to MI was restrained in CSE^−/−^ compared to WT mice (*p* < 0.001), while for *Akt1* expression, there was no significant difference between genotypes.

### 2.3. Pathway Analysis

To gain insight into the complex relationships between the signaling pathways activated in response to MI, to GYY4137 treatment and to CSE gene deletion, we used Qiagen Ingenuity Pathway Analysis (IPA). Comparing each experimental condition separately, the two most significant disease and function gene networks identified by IPA were annotated as Cardiovascular System Development and Function and Cardiac Necrosis/Cell Death, and these two networks were merged to create a single network incorporating the majority of genes examined (Figure 4 and Figure 5 for WT and CSE^−/−^ respectively). These gene networks highlight other signaling molecules within the hypertrophy, remodeling, fibrosis, antioxidant and apoptosis pathways that are functionally related to the genes we measured. Figure 4A illustrates the levels of expression with infarction in WT mice, and indicates strong activation in response to MI (red) of all network nodes increasing in concert, with the exception of *Agt* (with a functional connection to HDL). By contrast, Figure 4B compares the response to MI with GYY4137 administration to that of infarct alone in WT mice, where every gene node in these networks is green, indicating coordinated decreased expression. The same gene networks in CSE^−/−^ mice are shown in Figure 5, with MI associated with a more modest activation (less red) of the gene network (Figure 5A), with decreased expression of *Casp3* with MI in CSE^−/−^ mice (consistent with CSE gene deletion leading to a constrained response to MI of apoptotic signaling). Again, GYY4137 administration post MI (Figure 5B) had the effect of decreasing expression of every gene node in these networks except *MiR21a*. The comparison of WT and CSE^−/−^ mice is shown in Appendix A, indicating the influence of CSE gene depletion on expression across the same gene networks. In non-infarcted mice (Appendix A), expression levels of most genes were lower in CSE^−/−^ mice except for *MiR21a*, *Nppa*, *Ace*, *Agt*, and *Agtr1*. After infarct (Appendix A), the gene expression levels were also generally lower in CSE^−/−^ than WT mice, with only *MiR21a* and *Agt* expressed at higher levels in CSE^−/−^mice. 

### 2.4. Kidney Gene Expression 

Relative levels of expression of eight genes associated with remodeling and robustly expressed in kidney are shown in Figure 6 (for complete data see Appendix A). In contrast to findings in the heart, administration of GYY4137 drug alone in WT sham mice was associated with robust reductions in expression of *Nppa* (*p* = 0.005), *Tgfb1* (*p* = 0.003), *Col1a1* (*p* = 0.011), *Agtr1* (*p* = 0.006), and *Ace* (*p* = 0.004), but not in expression of *Agt*. Consistent with cardiac expression, infarct in WT mice resulted in elevated levels of kidney expression of *Nppa* (*p* = 0.018), *Tgfb1* (p = 0.004), and *Col1a1* (*p* = 0.004), but not significantly for *Agtr1*, *Ace*, or *Agt* expression. Again, the post-MI elevated levels of kidney gene expression in WT mice were restored to control by GYY4137 administration for *Tgfb1* (*p* < 0.001), *Col1a1* (*p* < 0.001), and for *Ace* (*p* = 0.008). In WT mice, kidney expression levels of the apoptosis genes, *Casp3* and *Akt1*, were not significantly altered either by infarct or by GYY4137 administration.

In CSE^−/−^ mice, expression levels did not change significantly with any treatment including infarction, in any of the remodeling genes, *Nppa*, *Tgfb1*, *Col1a1*, *Agtr*, *Agt*, *Ace*, *Casp3*, and *Akt1.* Comparing the genotypes, kidney *Nppa* expression was lower in CSE^−/−^ mice overall compared to WT, especially after infarction (*p* = 0.034 between genotypes). However, *Agt* expression levels were higher with GYY4137 treatment in CSE^−/−^ mice compared to WT in both sham (*p* = 0.005) and post-MI groups (*p* = 0.001). In contrast to the heart, *Casp3* expression levels in kidney were elevated in CSE^−/−^ mice (*p* = 0.003 between genotypes overall), especially post-MI, so that *Casp3* expression in the CSE^−/−^ infarct group was higher than in WT infarct (*p* = 0.015). Kidney expression of other genes, *Tgfb1*, *Ace*, *Col1a1*, *Agtr1*, and *Akt1*, was not different in CSE^−/−^ mice compared to WT.

## 3. Discussion

The primary aim of this study was to evaluate the utility of H_2_S therapy in ameliorating cardiac remodeling post-infarction in a mouse model. We found GYY4137 preserved cardiac function when administered from 2 h post-MI, a feasible clinical therapeutic time window after diagnosis of patients with MI. We demonstrated that daily GYY4137 administration from 2 h after the induction of MI and continued over a period of 28 days post-MI elicited sustained amelioration of the adverse effects of myocardial ischemia, including a reduction of infarct area and almost full recovery of cardiac function at the end of study. These functional benefits from GYY4137 were paralleled by restoration to near-normal ventricular expression levels for a host of genes associated with cardiac remodeling, including those mediating cardiac hypertrophy, fibrosis, antioxidant response, and apoptosis. While multiple signaling molecules have been associated with H_2_S, gene network analysis of representative genes from these pathways indicated that these functional networks converge and behave in a coordinated manner, being largely up-regulated with ischemia, universally down-regulated with GYY4137 treatment post-MI, and having generally reduced expression with endogenous CSE gene deletion (expression of *Mir21a*, *Nppa* and the genes of the renin-angiotensin system going against this trend in CSE^−/−^ mice). The overall effects of GYY4137 administration and of CSE gene deletion on cardiac function, and gene expression are summarized in the schematic are summarized in the schematic Figure 7.

A second objective of the study was to examine the role of endogenous CSE in the response to MI using CSE^−/−^ mice. Surprisingly, we saw no difference in survival post-MI between WT and CSE^−/−^ mice, and few differences between the genotypes in most measures of cardiac function or ventricular gene expression, either in non-infarcted control mice or in response to MI. Exceptions were higher blood pressures in CSE^−/−^ mice compared to WT (measured here under light anesthesia, noting that the this strain of CSE^−/−^ mice is reported to be normotensive when conscious [27]) We also observed markedly higher atrial *Mir21a* expression levels and slightly lower ventricular expression of *Casp3*, *Ace*, and *Nrf2* in CSE^−/−^ mice compared to their WT comparator groups, especially after infarction. In kidney, expression levels of most genes were lower in CSE^−/−^ mice at baseline compared to WT and changed little in response to GYY4137 treatment or infarction. Again, *Casp3* was the exception, with elevated kidney expression levels overall in CSE^−/−^ mice, especially post-MI. The gene network analyses reinforce the relatively modest effects of CSE gene deletion on the responses to either infarct or GYY4137 administration.

The higher blood pressures we observed in lightly-anesthetized CSE^−/−^ mice is in keeping with the actions of H_2_S to reduce vascular tone, and with reports that CSE^−/−^ mice develop elevated blood pressures with age [21]. However, the lack of marked differences between genotypes at 28 days post-MI was unexpected in light of the report by Kondo et al. [14] that CSE^−/−^ mice displayed significantly greater cardiac dilatation and dysfunction in response to transverse aortic constriction (TAC). This response was mitigated by administration of a H_2_S donor (12 weeks of SG-1002) and also in cardiac-specific, CSE over-expressing transgenic mice [14]. Cardiac-specific CSE over-expression has also been shown to reduce the extent of injury in response to myocardial ischemia-reperfusion [23]. Moreover, Miao et al. [22] found greater early (up to 8 days) post-MI pathological remodeling and dysfunction post-MI in CSE^−/−^ mice, which was ameliorated by NaHS administration acting via promotion of M2 macrophage polarization, achieved by accelerating internalization of integrin β1 and activating the downstream Src-FAK/Pyk2-Rac pathway [28]. Although this strain of CSE^−/−^ mice lack CSE gene expression across multiple tissues [27], it is possible that some H_2_S may be generated via alternative enzyme pathways involving CBS or 3-MST. It is possible the relatively normal response to MI in CSE^−/−^ mice reported here may result from a combination of opposing effects; reduced H_2_S-driven inhibition of cardiac remodeling but also a decreased anti-oxidant/inflammatory response to MI (lower *Ace*, *Nrf2* expression) and reduced apoptotic stimulus (lower *Casp3* expression). The strain of CSE^−/−^ mice used in the current study has been previously shown to have a reduced inflammatory response [29]. However, our findings suggest that CSE-derived generation of H_2_S is not obligatory to the normal response to MI, and CSE gene deletion does not lead to marked exacerbation of long-term cardiac remodeling compared to WT mice.

Our observation of upregulation of atrial *Mir21a* expression in CSE^−/−^ mice appears to counter a prior report that the H_2_S donor, Na_2_S induces *MiR21a* expression, and these authors proposed *Mir21a* mediates some of the cardioprotective effects of H_2_S [15]. Conversely, *MiR21a* has been associated with greater atrial fibrosis and collagen content in response to MI [16,30]. In the current study, we observed no difference in ventricular collagen expression in CSE^−/−^ compared to WT mice. To our knowledge, there have been no prior reports of CSE deletion altering *Casp3* expression, but we observed gene expression of this apoptotic protein to be lower in ventricle, whilst expression was higher in kidney tissue. Moreover, the gene pathway analysis in CSE^−/−^ mice as shown in Figure 5 demonstrated that GYY4137 administration post-MI decreased expression of every gene node in these networks except *MiR21a.* This indicates that administration of an exogenous H_2_S donor compound, GYY4137, while dampening the activation of most remodeling genes, was unable to reverse the elevated *MiR21a* associated with the absence of endogenous CSE.

We cannot determine whether the prolonged beneficial effects of GYY4137 post-MI are due to decreased drive towards remodeling (including hypertrophy and fibrosis) or to decreased early cardiac injury. However, it is intriguing that in WT mice, changes of kidney gene expression post-MI mirrored changes of ventricular gene expression, while kidney gene expression in CSE^−/−^ mice tended to be lower at baseline and not to respond to either GYY4137 treatment or infarct, despite a similar degree of cardiac injury in both genotypes. Thus, in the kidney, CSE depletion appears to have diminished the ability of hypertrophic signaling pathways to respond to the injury. These findings are suggestive that GYY4137 influenced the activation of remodeling gene signaling in a tissue-specific manner, rather than the response of signaling pathways post-MI across tissues being simply proportionate to the degree of cardiac ischemic damage. 

Past studies demonstrating beneficial effects of exogenous H_2_S post-MI have generally incorporated administration of sodium sulfide (Na_2_S) [31] or sodium hydrosulfide (NaHS) [32,33]. These substances are capable of releasing large amounts of H_2_S over a short duration (seconds), but may not replicate physiological concentrations within tissues. GYY4137 has the advantage of the slow release of H_2_S. Using an equivalent dose to our study, Lilyana et al. measured a 2.8-fold increase in plasma H_2_S levels in rats, this increase being relatively consistent at 2 and 7 days of treatment [3]. These authors demonstrated in this rat MI model that GYY4137 administration over a 7-day period attenuated early adverse cardiac remodeling, and concluded that this may have been moderated by activation of the cardiac hormone ANP, with increased *Nppa* (but not *Nppb*) expression by 2 days post-MI [3]. A study of GYY4137 administration over a 4-week timeline was conducted in spontaneously hypertensive rats (SHR) by Meng et al. [33], in which GYY4137 was found to elicit a dose-dependent reduction of myocardial fibrosis in the SHR. In cultured cardiac fibroblasts, GYY4137 also inhibited collagen synthesis and proliferation induced by angiotensin II [33]. The beneficial reduction of cardiac hypertrophy and fibrosis observed in the current study may be mediated in part by inhibiting activation of the cardiac renin-angiotensin system post-MI. 

We saw little effect of GYY4137 alone in non-infarcted mice apart from a slight increase in HW/BW and a small decrease in fractional shortening in WT. It was only after MI that we observed GYY4137 clearly reversed the elevated expression levels of most genes assessed. Consistent with our findings, beneficial effects reported for H_2_S include not only inhibition of hypertrophy and fibrosis, but also vasodilation, inhibition of leukocyte-endothelial cell interactions, antioxidant effects, anti-apoptotic effects, increased angiogenesis, and modulation of mitochondrial respiration [1]. Although NO signaling was not assessed in our study, these effects could have been mediated by H_2_S activation of NO pathways, since interactions between H_2_S and NO have been widely documented [9,13,14].

Limitations of this study include the fact that gene expression was only assessed at the end of study after 28 days, and hence we do not know the temporal profiles of gene expression over the intervening month after infarction. In addition, we were unable to normalize gene expression to endogenous reference genes due to insufficient ventricular tissue samples to assay the reference genes from all samples across the treatment groups.

In summary, we have demonstrated that in both WT and CSE-deficient mice, the H_2_S donor GYY4137 administered after onset of MI, preserved cardiac function and protected against adverse cardiac remodeling.

## 4. Materials and Methods

### 4.1. Myocardial Infarction

The study protocol was approved (reference no. C1/13, approval 25 February 2013) and performed in accordance with the guidelines of the Institutional Animal Care and Use Ethics Committee of the University of Otago, Christchurch, New Zealand (NZ). Sixty-four 12-week old male mice (*n* = 32 CSE^−/−^ and 32 WT, both on C57/BL6J background) randomly underwent either ligation of the left coronary artery to induce an MI (*n* = 32; 16 CSE^−/−^, 16 WT) or sham control surgeries (thoracotomy without coronary ligation *n* = 32; 16 CSE^−/−^, 16 WT) [34]. The generation of the CSE^−/−^ mice has been previously described [27]. The MI was induced as described previously [34,35,36]. In brief, a left lateral thoracotomy was performed under anesthesia induced by a single subcutaneous (SC) injection of 75 mg/kg ketamine (Ceva Ketamine, Ceva Animal Health Pty Ltd., Glenorie, NSW, Australia) and 1 mg/kg medetomidine hydrochloride (Domitor, Zoetis New Zealand Ltd., Auckland, NZ), with ventilation at 107 breaths/min; 1.5 mL tidal volume [34]. The pericardium was opened, and the left coronary artery ligated mid-way between the left atrium and the apex of the left ventricle. Successful MI was confirmed by blanching and dyskinesis of the distal myocardium [34]. Sham-operated control mice were treated similarly, but without ligation of the coronary artery. Anesthesia was then reversed by SC 1 mg/kg atipamezole (Antisedan, Orion Pharma, Espoo, Finland) and the mice were observed in a temperature-controlled environment until fully recovered, before being returned to the animal-holding facility where they were given standard chow and water ad libitum. Post-operative analgesia was provided (SC buprenorphine (Temgesic, Indivior UK Ltd., Slough, UK), 0.1 mg/kg/day) over the next 2 days [34].

### 4.2. Treatment Protocol

Infarcted and sham animals were randomly allocated to two groups to receive vehicle control (0.9% saline intraperitoneal (IP) administration; n = 6–8/group/genotype) or the hydrogen sulfide donor GYY4137 (morpholin-4-ium-methoxyphenylmorpholino-phosphinodithioate, gifted by Prof. Phil Moore, Pharmacology, National University of Singapore [26]), administered 50 mg/kg/day IP starting 2 h post-surgery, and continuing daily for 28 days (*n* = 5–10/group/genotype). This dose regimen has been shown in rats to increase plasma H_2_S levels 2.8-fold [3]. The treatment period was chosen as the time required to observe significant hypertrophic and fibrotic changes post-MI in this model, reflecting long-term remodeling rather than the immediate response to injury [34,35,36]. 

### 4.3. Echocardiography

Cardiac function was measured at 28 days by transthoracic echocardiography using an iE33 ultrasound machine (Philips Ultrasound, Bothell, WA, USA) with a 15-MHz linear transducer, as described previously [35,36]. The mice were placed in a shallow left lateral position under light inhalation anesthesia with 3.5% isofluorane (with a flow rate of 270 mL/min of oxygen) and maintained with 1.8% isofluorane (with a flow rate of 270 mL/min of oxygen) via nose cone. A 2-dimensional (2D) image obtained in the parasternal short axis view at the level close to the papillary muscles, and a 2D M-mode trace recorded across the anterior and posterior wall of the left ventricle. Indices reported here are fractional shortening (FS) and left ventricular ejection fraction (LVEF) [35,36]. Mean arterial blood pressure (MAP) was measured at the same time by a noninvasive computerized tail cuff system (ADInstruments, Dunedin, NZ).

### 4.4. Tissue Collection for Histology and RNA Isolation

At the end of the 28-day study period, the mice were weighed prior to being euthanized with an anesthetic overdose (Isoflurane inhalation (Attane™, Bayer New Zealand Ltd., Auckland, NZ)), followed by cervical dislocation, as previously described [34,35,36]. At sacrifice, the hearts were rapidly excised before being weighed for calculation of heart weight/body weight ratios (HW/BW), the hearts were photographed on a marked grid, the perimeter of the visible infarct traced and the area calculated with the use of ImageJ software (National Institutes of Health, Bethesda, MD, USA; version 1.43u) [35]. In addition, two hearts from each infarct group were reserved for histological examination, fixed in 10% formalin before being paraffin-embedded and sections (7 µm thick) cut from whole hearts for staining with Masson trichrome. To assess the extent of fibrosis post-MI, digital images were taken of the left ventricle (LV) at 4× objective magnification (representative sections shown in Figure 1) [35]. From the remaining mice, hearts and kidneys were processed for RNA isolation, with the ventricles dissected from the atria and all tissues immediately snap frozen in liquid nitrogen.

Total RNA was isolated from cardiac ventricle, atrium, and kidney tissue samples for quantitative real-time PCR (RTqPCR) analysis, as previously described [34,35,36]. Briefly, after automated grinding in a Retsch MM301 tissue mill (Retsch GmbH & Co KG, Haan, Germany) at 30 Hz for 10 min in 800 µL pre-chilled TRIzol™ (Invitrogen, Carlsbad, CA, USA), chloroform (160 µL) was added and samples were centrifuged at 12,000× *g* for 15 min at room temperature. The RNA-containing supernatant was purified by RNeasy Midi columns according to the manufacturer’s instructions (Qiagen, Hilden, Germany). For microRNA (miRNA) analysis from atrial tissue, total RNA was isolated using the Cell and Plant miRCURY RNA Isolation Kit (Qiagen kit) according to the manufacturer’s instructions. cDNA synthesis was performed from 2 μL total RNA with the universal cDNA kit (Qiagen) following the protocol of the supplier. Synthetic spike-in controls were added to each sample prior to RNA extraction (UniSp2, UniSp4, UniSp5) and cDNA synthesis (UniSp6, cel-39-3p) as provided in the Qiagen RNA Spike-in and Universal cDNA kit. 

### 4.5. Quantitative Real-Time PCR Analysis

For RTqPCR analysis of mRNA for genes of interest, cDNA was generated from 2.5 μg of ventricular total RNA as previously described [34]. The PCR reaction conditions were optimized with oligonucleotide primer sequences and PCR annealing temperatures for each gene studied (Appendix A). PCR reactions were performed in a total volume of 30 µL containing 1 µL cDNA, 0.4 mM primers, 1×PCR buffer, 0.2 mM dNTPs, 1.5 mM MgCl_2_, 10 nM SYBR Green 1 (Roche Diagnostics, Indianapolis, IN, USA), and 1 U TAQ-Ti DNA Polymerase (Fisher Biotec Australia, Wembley, WA, Australia). Once optimized, levels of mRNA expression were evaluated by quantitative RTqPCR in a Lightcycler 480 Real-Time PCR system (Roche Diagnostics Ltd., Rotkreuz, Switzerland) in duplicate using the following conditions: hotstart at 96 °C for 2 min, and then 30 cycles of denaturation at 94 °C for 30 s, annealing for 35 s at the gene-specific annealing temperature (Appendix A), and extension at 72 °C for 30 s, after which a melt curve was performed [34,35,36]. For miRNA analysis, RTqPCR assays were performed in triplicate on a LightCycler 480 Real-Time PCR instrument (Roche) using Qiagen chemistry, in 10 μL reaction volume with the assay detection limit set at a *C*t of 40 cycles.

Expression levels of mRNAs or miRNAs were converted to relative quantities with the highest value across groups set as 1. Normalization to endogenous reference genes was undertaken where tissue samples were available for atrium, ventricle, and kidney, to either three (*Snord68*, *5S rRNA*, *Rnu1a1* in atrium and kidney) or four (*Hprt*, 5S rRNA, *18S rRNA*, *Rnu1a1* in ventricle) endogenous reference genes, identified as the most stable combination from five candidates (including hsa-miR-93-5p) using geNorm software [37] (https://genorm.cmgg.be) and qbase+ software, version 3.1 (Biogazelle, Zwijnaarde, Belgium, www.qbaseplus.com). Because there was insufficient tissue to normalize all samples, and normalized data were highly correlated with raw expression data (all genes Pearson correlation coefficients were between 0.750 and 0.932, *p* < 0.0001, see Appendix A), non-normalized data are presented in the final analyses to maximize group sizes.

### 4.6. Pathway Analysis

To identify the functional gene networks altered in association with each specific experimental condition (myocardial infarction, GYY4137 treatment, and CSE gene deletion), gene networks and functional analyses were generated through the use of IPA (QIAGEN Inc., https://www.qiagenbio-informatics.com/products/ingenuity-pathway-analysis) [38]. For each gene, relative expression fold-change and *p*-value data were entered as pairwise comparisons between experimental groups (sham versus infarct, WT vehicle treated versus GYY4137 treated, WT versus CSE^−/−^ etc.). For each experimental condition, the most significant two unbiased disease and function annotation networks were merged to produce network diagrams of the overall effects of each condition on gene expression.

### 4.7. Statistical Analysis 

Two samples of ventricle were excluded due to poor efficiency of RNA extraction, leaving 51 samples in the analysis. Final numbers per group: WT sham vehicle *n* = 6, WT sham GYY4137 *n* = 3, WT infarct vehicle *n* = 6, WT infarct GYY4137 *n* = 10, CSE^−/−^ sham vehicle *n* = 7, CSE^−/−^ sham GYY4137 *n* = 7, CSE^−/−^ infarct vehicle *n* = 6, CSE^−/−^ infarct GYY4137 *n* = 8. All mRNA and miRNA expression data displayed skewed distributions and were natural-log transformed prior to statistical analysis. Log-transformed relative expression levels in each of the groups were compared using a univariate linear model in which genotype, infarct, and drug treatment were assessed as fixed factors, as well as any interaction between these factors, followed by pairwise comparisons to determine which groups differed from the rest. All statistical analyses were performed with SPSS statistics software and a *p*-value of 0.05 was taken to indicate statistical significance.

## 5. Conclusions

Administration of a slow release H_2_S donor, GYY4137, delivered from 2 h after the onset of MI, ameliorated adverse cardiac remodeling and was associated with preservation of cardiac function. In ventricular tissue, GYY4137 was also associated with suppression of the post-MI rise in expression of genes associated with cardiac remodeling, including those mediating cardiac hypertrophy, fibrosis, antioxidant response, and apoptosis. In kidney, expression levels of most genes changed little in response to either GYY4137 treatment or to infarction. These findings lead us to suggest that endogenous H_2_S signaling generated by CSE is not essential to the response to MI, while exogenous administration of an H_2_S donor elicits major beneficial effects in the long-term response to MI. Therefore, this study demonstrates that the use of H_2_S donor drugs is a potential therapeutic strategy, even when delivered after the onset of MI.

## Figures and Tables

**Figure 1 ijms-21-04284-f001:**
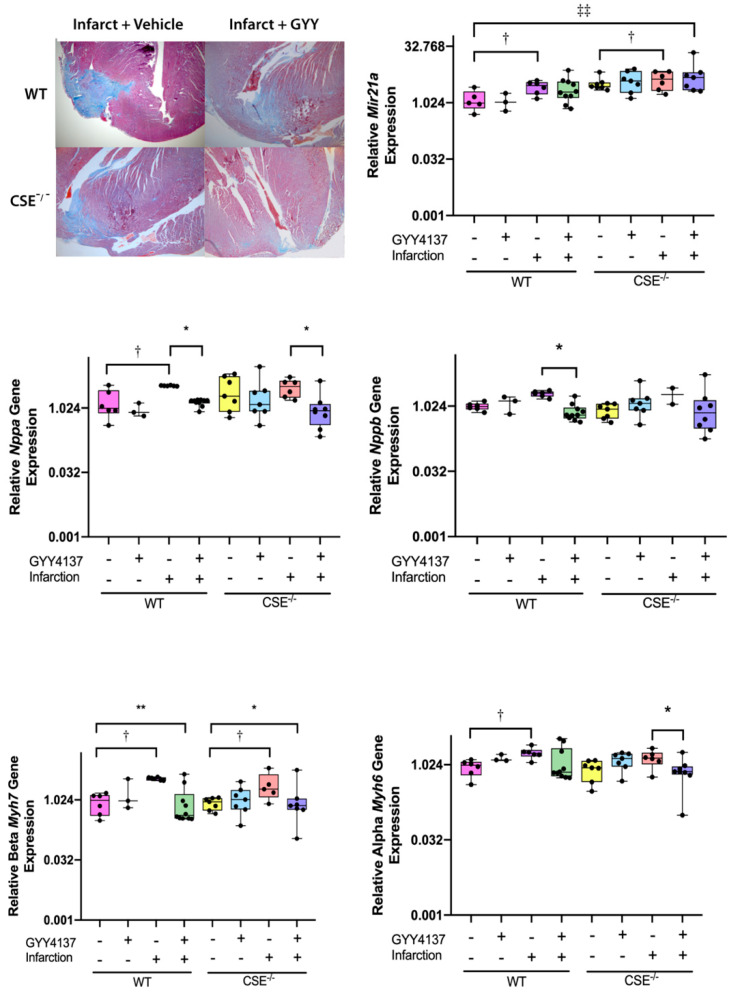
Expression of genes associated with cardiac hypertrophy at end of study (28 days). Top left panel shows representative histological sections of infarcted ventricle (photographed at 4X magnification) stained with Masson trichrome to indicate collagen (blue). Remaining panels are boxplots of gene expression of atrial *MiR21a* (top); ventricle ANP (*Nppa*) and BNP (*Nppb*) (center row); βMHC (*Myh7*) and αMHC (*Myh6*) (bottom row). All graphs are medians and Interquartile ranges, whiskers are 95% CIs. For complete data see Appendix A. * *p* < 0.05, ** *p* < 0.001 for GYY4137 vs. vehicle; † *p* < 0.05 for effect of sham vs. infarct; ‡‡ *p* < 0.001 for wild type vs. CSE knockout mice (CSE^−/−^).

**Figure 2 ijms-21-04284-f002:**
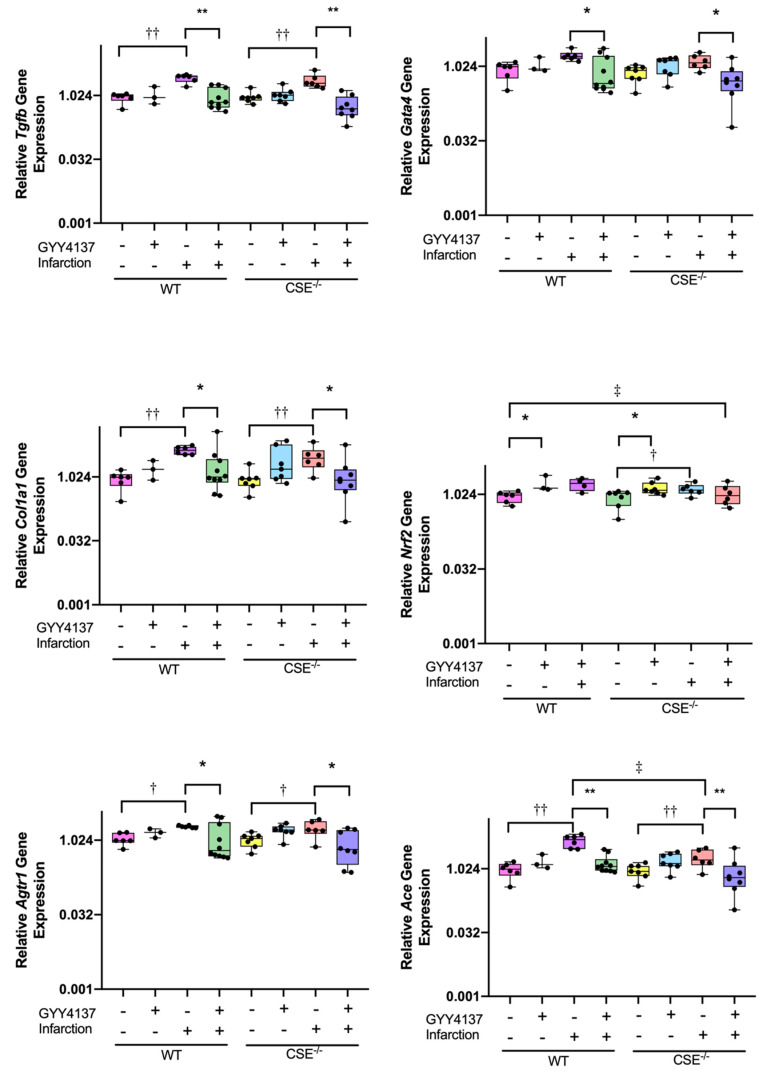
Expression of cardiac remodeling and fibrosis genes at end of study (28 days). Boxplots of ventricle tissue gene expression of TGFβ1 (*Tgfb1)* and GATA4 *(Gata4)* (top row); Collagen 1a1 (*Col1a1)*, and NRF (*Nrf2)* (center row); and: AT1R (*Agtr1*) and ACE (*Ace)* (bottom row). All graphs are medians and Interquartile ranges, whiskers are 95% CIs. For complete data, see Appendix A. * *p* < 0.05, ** *p* < 0.001 for GYY4137 vs. vehicle; † *p* < 0.05, †† *p* < 0.001 for effect of sham vs. infarct; ‡ *p* < 0.05 for wild type vs. CSE^−/−^.

**Figure 3 ijms-21-04284-f003:**
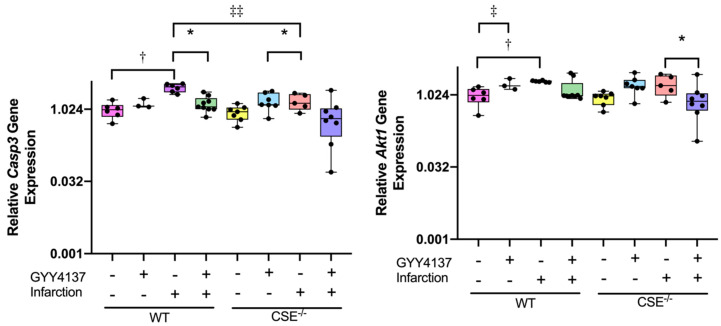
Expression of apoptosis and antioxidant genes in cardiac tissue at end of study (28 days). Boxplots of ventricle tissue gene expression of Caspase 3 (*Casp3)* and Akt1 (*Akt1*). All graphs are medians and interquartile ranges, whiskers are 95% CIs. For complete data, see Appendix A. * *p* < 0.05 for GYY4137 vs. vehicle; † *p* < 0.05 for effect of sham vs. infarct; ‡ *p* < 0.05, ‡‡ *p* < 0.001 for wild type vs. CSE^−/−^.

**Figure 4 ijms-21-04284-f004:**
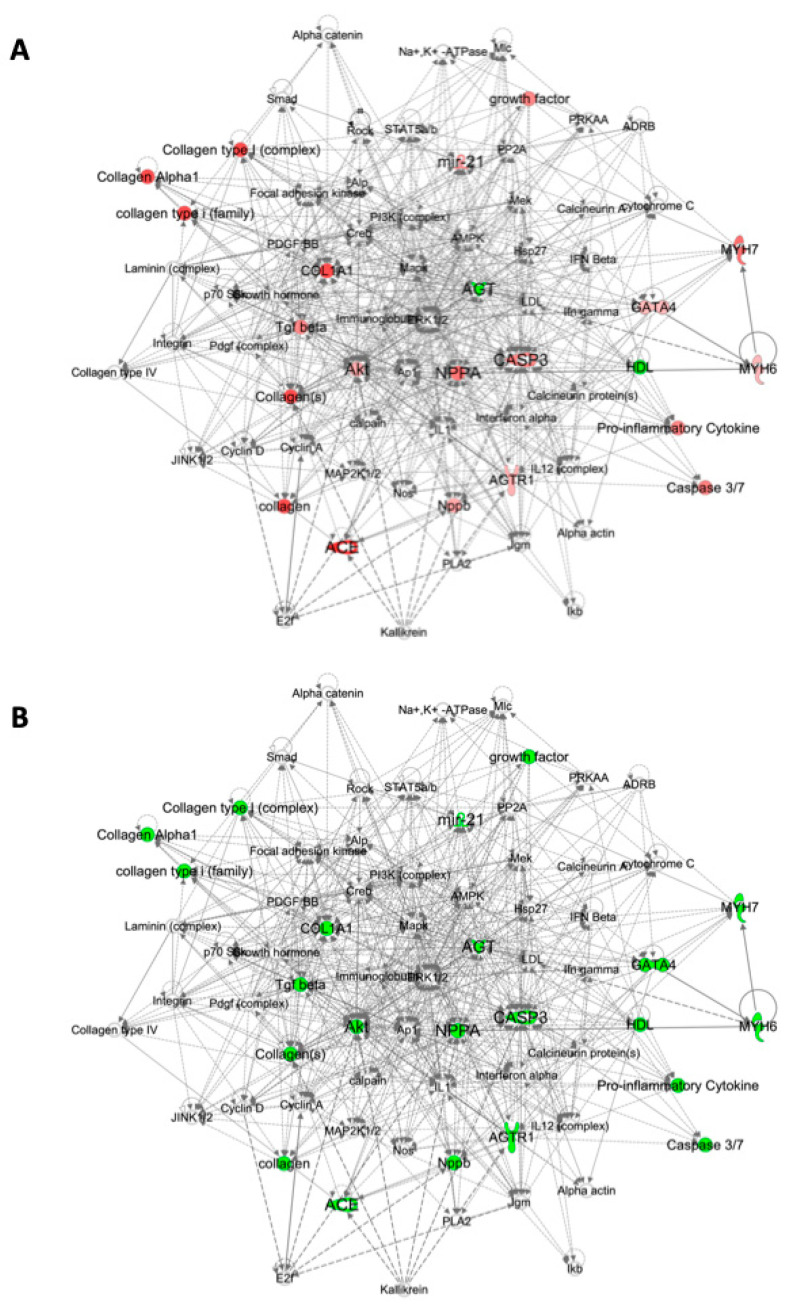
Cardiac gene networks in wild-type (WT) mice comparing infarct without and with GYY4137. Gene networks illustrating direct and indirect functional relationships between cardiac genes in WT mice comparing: (**A**) WT sham vehicle vs. WT infarct vehicle; (**B**) WT infarct vehicle vs. WT infarct GYY. Highlighted genes have higher (red) or lower (green) expression in the second treatment group compared to the first group in each network. The color intensity correlates with the magnitude of the fold-change in the expression between treatment groups. Gene networks were generated through the use of Ingenuity Pathway Analysis (IPA) (QIAGEN Inc., https://www.qiagenbio-informatics.com/products/ingenuity-pathway-analysis).

**Figure 5 ijms-21-04284-f005:**
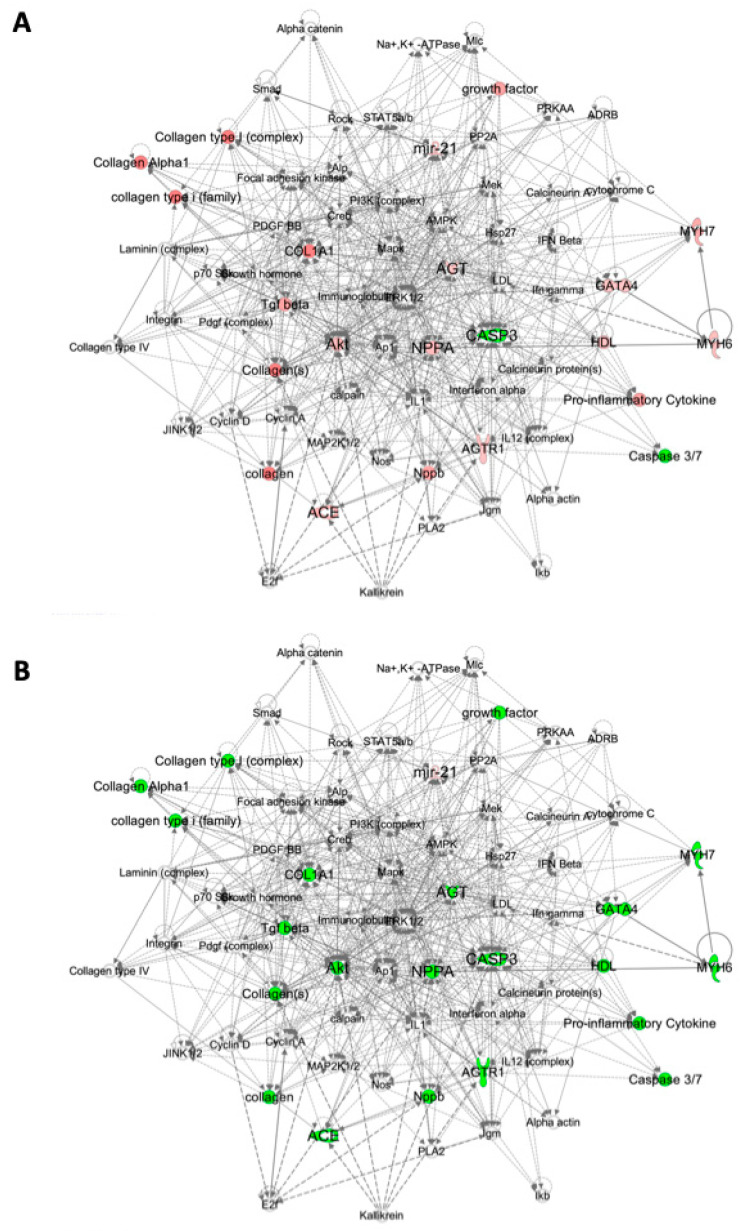
Cardiac gene networks in CSE^−/−^ mice comparing infarct without and with GYY4137. Gene networks illustrating direct and indirect functional relationships between cardiac genes in CSE^−/−^ mice: (**A**) CSE^−/−^ sham vehicle vs. CSE^−/−^ infarct vehicle; (**B**) CSE^−/−^ infarct vehicle vs. CSE^−/−^ infarct GYY. Highlighted genes have higher (red) or lower (green) expression in the second treatment group compared to the first group in each network. The color intensity correlates with the magnitude of the fold-change in the expression between treatment groups. Gene networks were generated through the use of IPA. (QIAGEN Inc., https://www.qiagenbio-informatics.com/products/ingenuity-pathway-analysis).

**Figure 6 ijms-21-04284-f006:**
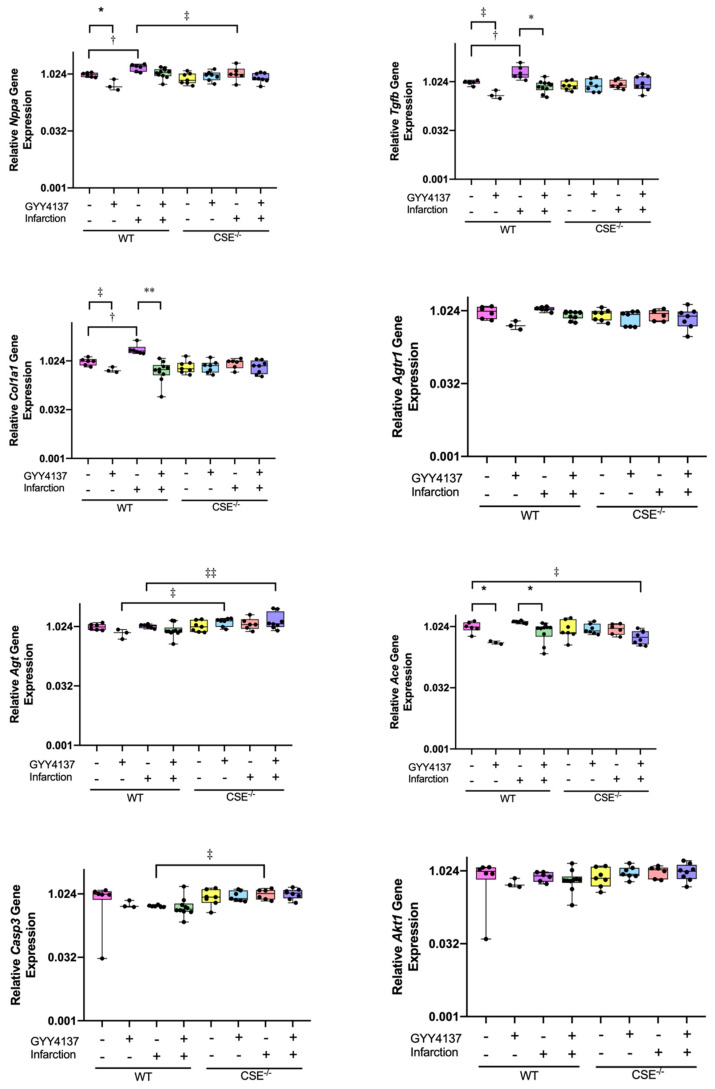
Expression in kidney tissue of genes associated with remodeling at end of study (28 days). Boxplots of kidney tissue gene expression of ANP (*Nppa*) and TGFβ1 (*Tgfb1)* (top row); collagen 1a1 (*Col1a1)* and AT1R (*Agtr1*) (2nd row); AGT (*Agt*) and ACE (*Ace*) (3rd row); caspase 3 (*Casp3)* and Akt1 (*Akt1*) (bottom row). All graphs are medians and interquartile ranges, whiskers are 95% CIs. * *p* < 0.05, ** *p* < 0.001 for GYY4137 vs. vehicle; † *p* < 0.05 for effect of sham vs. infarct; ‡ *p* < 0.05, ‡‡ *p* < 0.001 for wild type vs. CSE^–/–^.

**Figure 7 ijms-21-04284-f007:**
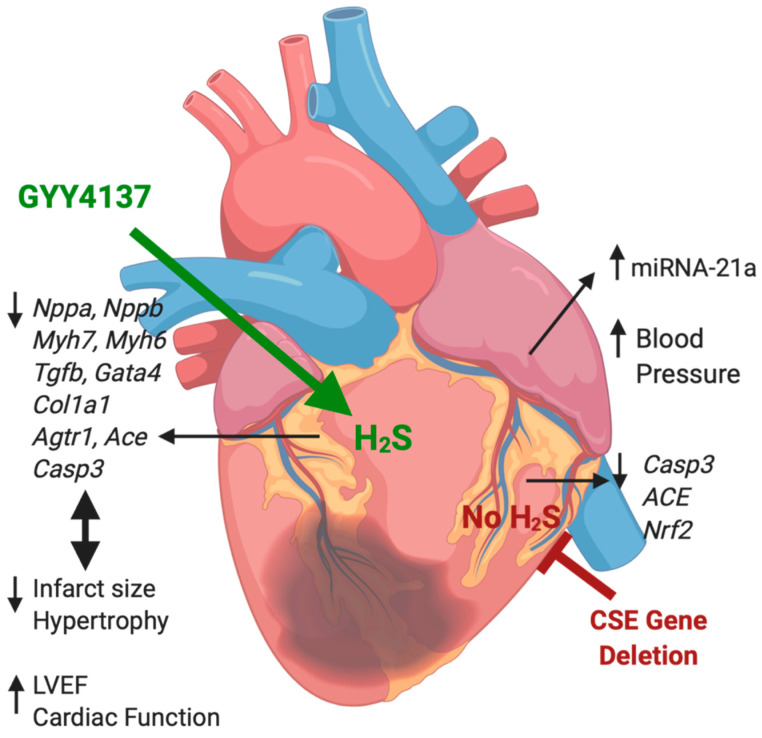
Cardiac effects of GYY4137 and CSE gene deletion. The cardiac effects of GYY4137 administration after myocardial infarction (left of figure), include marked reduction of the post-infarct elevation in ventricular gene expression associated with preservation of cardiac function. The effects of CSE gene deletion in CSE^−/−^ mice (right of figure), include higher blood pressure, increased atrial *Mir21a* expression in both sham and post-infarct mice, and lower post-infarct expression of specific genes in ventricular tissue (Drawing created with BioRender.com).

**Table 1 ijms-21-04284-t001:** Cardiac function and echocardiography indices at 28-day end of study (mean ± sem).

	WT Sham Vehicle	WT Sham GYY4137	WT Infarct Vehicle	WT Infarct GYY4137	CSE KO Sham Vehicle	CSE KO Sham GYY4137	CSE KO Infarct Vehicle	CSE KO Infarct GYY4137
MAP mmHg	73.5 ± 3.74	68.8 ± 4.8	66.3 ± 3.4	67.5 ± 2.6	94.8 ± 3.2 ^‡‡^	101.4 ± 3.2 ^‡‡^	86.4 ± 3.4 ^‡^	76.12 ± 3.4 *^,‡^
Infarct mm^2^	-	-	5.80 ± 0.71	1.50 ± 0.58 *	-	-	7.33 ± 2.03	1.75 ± 0.37 *
HW/BW	5.39 ± 0.17	5.73 ± 0.23	6.81 ± 0.17 ^††^	5.79 ± 0.13 **	5.06 ± 0.15	5.71 ± 0.15	6.76 ± 0.17 ^††^	5.91 ± 0.14 **
LVEF%	74.7 ± 1.28	70.1 ± 1.80	46.8 ± 1.28 ^††^	64.2 ± 0.99 **	72.3 ± 1.18	68.9 ± 1.18	40.8 ± 1.28 ^††,‡^	62.7 ± 1.18 **
FS%	37.5 ± 0.89	32.0 ± 1.26	17.67 ± 0.89 ^††^	29.0 ± 0.69 **	35.57 ± 0.82	39.57 ± 0.82	15.83 ± 0.89 ^††^	30.13 ± 0.77 **
LV mass g	0.71 ± 0.01	0.68 ± 0.01	0.74 ± 0.03	0.69 ± 0.03	0.69 ± 0.01	0.73 ± 0.02	0.69 ± 0.02	0.69 ± 0.01
LVPWs mm	1.36 ± 0.08	1.18 ± 0.07	0.99 ± 0.14 ^†^	1.28 ± 0.14	1.06 ± 0.80 ^‡^	1.49 ± 0.08	0.97 ± 0.09 ^††^	1.08 ± 0.07
IVS%	0.96 ± 0.04	1.01 ± 0.04	1.31 ± 0.08 ^††^	0.80 ± 0.08 **	0.96 ± 0.04	0.81 ± 0.04	0.87 ± 0.05 ^‡‡^	0.84 ± 0.04
EVS mL	0.05 ± 0.02	0.05 ± 0.02	0.05 ± 0.03	0.06 ± 0.03	0.03 ± 0.02	0.07 ± 0.02	0.04 ± 0.02	0.05 ± 0.02
EDV mL	0.12 ± 0.08	0.07 ± 0.07	0.13 ± 0.15	0.13 ± 0.15	0.22 ± 0.08	0.19 ± 0.08	0.26 ± 0.09	0.20 ± 0.07

* *p* < 0.05, ** *p* < 0.001 for GYY4137 vs. vehicle; ^†^
*p* < 0.05, ^††^
*p* < 0.001 for effect of sham vs. infarct; ^‡^
*p* < 0.05, ^‡‡^
*p* < 0.001 for wild type vs. CSE^−/−^. Abbreviations: MAP—mean arterial pressure; Infarct—Infarct size (area of infarct scar after 28 days); HW/BW—Heart weight to body weight ratio; LVEF—left ventricular ejection fraction; FS—fractional shortening; LVPWs—left ventricular posterior wall thickness (systolic); IVS—interventricular septum (systolic); EVS—end-systolic volume; EDV—end-diastolic volume.

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
