# Peer review of "Hydrogen Sulfide Treatment Improves Post-Infarct Remodeling and Long-Term Cardiac Function in CSE Knockout and Wild-Type Mice"

_ijms, 2020, doi:10.3390/ijms21124284_

Round 1

Reviewer 1 Report

This is an interesting and well-presented study on the effect of hydrogen sulphide on long term effects post-MI.

Major comments:

(1) Please provide information on other ECHO parameters, which will have been obtained together with EF and FS? EF and FS give very similar information, but volumes and masses would be informative. A table of all ECHO parameters would be best. Especially considering that the aim of the study is long term remodelling, readouts relating to LV dilation are important.

(2) Please include the heart photographs which were used to measure infarct size in the supplement. The described way of obtaining infarct size appears rather crude and including the pictures would increase confidence in this crucial measure.

(3) Please also include the analysis performed to assess the suitability of using raw expression for qPCR results. The materials and methods section describes a sophisticated way of normalising against several reference genes, but then the decision was made to use raw results due to insufficient tissue. This should be justified by showing the supporting data referred to in the supplement.

(4) The difference between anticipated and observed effects of global CSE deletion (as opposed to previously published cardiac-specific CSE deletion) are interesting and warrant a more detailed discussion. Especially the difference between WT and KO effects in the kidney.

Minor/editorial comments:

(1) Abstract: A brief introduction to H2S and CSE would be helpful.

(2) Line 110: please state what the values are; mean or median? CI?

(3) Please start axes at 0 to allow the reader a better estimate of true differences. Splitting axes is fine.

(4) Please refrain from describing statistically non-significant results as ‘having a trend/tended to change/tended not to change’ or similar. Group sizes should be based on power calculations and results are statistically significant or not as per defined p value. Limitations of p value-based analysis are accepted. The results remain and biological significance may differ from statistical significance, but this needs to be described in a different way.

(5) The results section describing qPCR results is extensive and I advise shortening substantially to improve readability. Also the pathway analysis description is not very clear at the moment. Relevance and conclusion drawn from it are hard to understand.

Author Response

Response to Reviewer 1

  1. Please provide information on other ECHO parameters; volumes and masses would be informative:

This is a valuable suggestion and we have now included additional echocardiography indices with comparisons of masses, wall thickness, and volumes. As requested by Reviewer 1, we have included these in a new Table1:Cardiac Function and Echocardiology Indices at 28-day End of Study. To avoid duplication of results, we have deleted the original Figure 1, which contained similar information in graphical form, and replaced this with the new Table 1. All subsequent figure numbers have been amended in the text and figures themselves to account for the removal of the previous Figure 1.

  1. Please include the heart photographs which were used to measure infarct size:

We are unable to address this request, as we did not archive the photographs of the hearts once the infarct area at end of study was calculated and documented. The Reviewer considers our technique to be a rather crude method for estimating infarct size, but we respectfully mention that we have previously published this method in multiple publications (see previous publications: Ellmers et al., Endocrinology 2008149(11):5828;  Ellmers et al., J Cardiovasc Pharmacol 201565(3):269; Ellmers et al.,J Cardiac Fail 201622(1):64).

  1. Please include the analysis performed to assess the suitability of using raw expression for qPCR results:

As suggested, we have now added a Supplementary Table 4 showing the Pearson Correlation analysis between the raw and the normalised ventricular expression data. This demonstrates that for all genes Pearson correlation coefficients were between 0.0.750 and 0.932 (p<0.0001). The Table is referred to in the Methods at line 502-503. Since the Results precede the Methods in the manuscript, numbering of the original Supplementary Tables 2 and 3 has been adjusted to 1 and 2, and references to these in the text have been changed accordingly.

  1. The differences between anticipated and observed effects of global CSE deletion (as opposed to previously-published cardiac-specific CSE deletion) are interesting and warrant a more detailed discussion:

We have performed an extensive literature search for papers reporting cardiac-specific CSE gene deletion, and were unable to find any publications on this topic (although there are papers on cardiac-specific CSE over-expressing transgenic mice). If the Reviewer is able to provide citations, we would be happy to expand the Discussion on the topic of cardiac-specific CSE deletion.  We have now included in the Discussion the mention of two studies of cardiac-specific CSE overexpression (lines 336 -  337), which emulate the effects of exogenous H2S administration (Elrod et al., H2S is a physiological vasorelaxant: Hypertension in mice with deletion of CSE. PNAS2007; Kondo et al., H2S protects against pressure overload-induced heart failure via upregulation of NO synthase Circulation2013).

Minor /editorial Comments

  1. Abstract: A brief introduction to H2S and CSE would be helpful:

We have added another sentence to the start of the Abstract (lines 19-22), as follows: “Hydrogen sulphide (H2S) is recognised as an endogenous gaseous signalling molecule generated by cystathionine g-lyase (CSE)in cardiovascular tissues. H2S up-regulation has been shown to reduce ischaemic injury, and H2S donors are cardioprotective in rodent models when administered concurrent with myocardial ischaemia.”

  1. Line 110: Please state what the values are, mean or median? CI?

These data in the text (previously line 110) were means ± sem, but the data are now included in the new Table 1 as requested by this Reviewer and have been removed from the text to avoid duplication. Table 1 now provides a clear statement that all the data therein is expressed as means ± sem.

  1. Please start axes at 0 to allow the reader to get a better idea of the differences . Splitting axes is fine:

All graphs depicting gene expression have now been re-drawn to allow the reader a more consistent estimate of the differences. The Reviewer requested the axes start at zero, but because the graphs are log scale, zero is not mathematically possible. We have therefor re-drawn the graphs with the y axis consistently starting at 0.001.

  1. Please refrain from describing statistically non-significant results as “having a trend...”:

All reference to trends in the text have been removed.

  1. The Results section describing qPCR results is extensive and I advise shortening substantially:

Results section describing qPCR has been shortened substantially, as requested, reducing the number of lines of text from 227 to 188 lines. This has been achieved by consolidating the findings for expression of several genes with similar functions and also by removing the inclusion of data within the text (as it is available in the Supplemental Tables). We hope the Reviewer will agree that this makes the Results more succinct and easier to read.

Also the pathway analysis section is not very clear and the conclusions drawn from it are hard to understand:

We have expanded this section of Results to clarify the interpretation of the pathway analysis at line 226, in lines 232 to 234, at line 241 and throughout this section. We have also added lines 309 to 314 to the Discussion.

Reviewer 2 Report

The study by L. Ellmers and co-authors is devoted to the role of H2S in cardiac ischemic adaptation and prevention of post-MI maladaptive remodeling in mouse model. The authors presented an elegant study which includes a proper number of controls and reference groups and followed the effects of naïve MI remodeling and drug-treated remodeling after 28 days of MI. Additionally, the comparison was made between WT and genetically modified mice lacking CSE expression. The authors demonstrated that delayed H2S administration after 2 hours post MI protects from maladaptive gene expression profile, hypertrophy and fibrosis and has also a beneficial effect on Echo parameters.  Apart from studying the cardiac-restricted processes, kidney tissue analysis also supported the protective role of H2S in post-MI events. Surprisingly, authors were not able to detect a prominent impact of CSE depletion on cardiac and kidney function 28 days after MI and underlined the tissue -specific effects of CSE enzyme during ischemia-reperfustion injury.

 The study is well-planned and written, there are sufficient number of groups studied with a proper number of animals which allows to make a correct statistical analysis. The illustration material is clear and well described, the statistical analysis is presented in the proper form on all graphics. All relevant but not primary important information in illustrated in supplemental materials.

 There are several minor points which needs to be added or described, they can give an additional information and clear up some moments:

  1. In Figure 2 Mir21 expression is not compared between WT and CSE- mice at base line. Please, comment it in the test or as ns (?) on the figure. It might be important for the readers.
  2. In section 2.2 in data description and also in Figure legends I recommend to add the information on time point assessed (28 days). It clearly appears only in MM section which is the last one and make it difficult to follow.
  3. Please, explain the choice of RNA expression markers, such as GATA4, NPPb or NRF2. What is a theoretical background behind these genes? For example, why Gata4 was chosen? There are other embryonic program markers in the heart. These genes need more attention in the introduction and discussion.
  4. Line 190- “was” twice
  5. It is not quite clear why the limitation for expression normalisation was in tissue size. Usually the small part of cardiac tissue gives the opportunity even for RNA seq. were there any technical problems?

 In general, study is well planned and performed and presented in clear detailed manner. I recommend it for a publication after minor revision.

Author Response

Response to Reviewer 2

The Reviewer raises the following points.

  1. In Figure 2 Mir21 is not compared between WT and CSE–/–mice at baseline. Please comment on this in the text or add ns to the figure:

In the manuscript text, a statement that there was “a significant pairwise difference between genotypes noted even in the baseline condition of non-infarcted mice without drug administration (p=0.002)”has been added in lines 155-156. However, to avoid cluttering the graphs, only statistically significant difference between the genotype groups overall is indicated on the graphs.

  1. In section 2.2 in data description and also in Figure legends, I recommend to add information on the time point assessed (28 days):

This information has now been added to the Results section (line 111), to Table 1 and to the legends to Figures 1, 2, 3, and 6.

  1. Please explain the choice of RNA expression markers, such as GATA4, NPPB, NRF2. What is the theoretical background behind these genes? These genes need more attention in the introduction and in the Discussion:

Additional discussion of the rationale for selecting these genes, as representatives of pathways reported to be involved in H2S signalling, has been added to both the Introduction (lines 98 to 102) and in the Discussion (lines 309 to 314).

  1. Line 190 – “was” twice:

The typographical error has now been corrected. Thank you.

  1. It is not clear why the limitation for expression normalisation was in tissue size. Usually the small part of cardiac tissue gives the opportunity even for RNA seq. Were there any technical problems?

We intentionally opted for a qPCR approach rather than a discovery method such as RNASeq, as we would have needed a very large sample size to provide statistical power for an RNASeq approach. The entire left ventricle was used for RNA extraction, which was sufficient for RT-qPCR analysis of the twelve selected genes for the pathways previously associated with H2S function in cardiac tissue. Unfortunately, for some samples the twelve gene expression assays used up the RNA, leaving smaller group sizes for the reference gene expression assays (as can be seen in the new Supplementary Table 4, comparing sample numbers per group in the raw data versus the normalised data). While we considered analysing and presenting our normalised data, that would have involved reporting gene expression in reduced group sizes (where reference gene expression was available), reducing statistical power of the study. Because the raw and normalised data were so highly correlated, we have analysed and presented the raw data in the manuscript.

Round 2

Reviewer 1 Report

I thank the authors for their thorough response to my comments. All concerns have been adressed.